# Randomized Controlled Trial Comparing the Efficacy of Sustained-Release Formula of Mosapride-Plus-Esomeprazole Combination Therapy to Esomeprazole Monotherapy in Patients with Gastroesophageal Reflux Disease

**DOI:** 10.3390/jcm11071965

**Published:** 2022-04-01

**Authors:** Hye Kyung Jeon, Gwang Ha Kim, Moon Won Lee, Dong Chan Joo, Bong Eun Lee

**Affiliations:** 1Department of Internal Medicine, Pusan National University College of Medicine, Busan 49241, Korea; kyung3842@hanmail.net (H.K.J.); neofaceoff@hanmail.net (M.W.L.); oceanose@korea.ac.kr (D.C.J.); bongsul@hanmail.net (B.E.L.); 2Biomedical Research Institute, Pusan National University Hospital, Busan 49241, Korea

**Keywords:** gastroesophageal reflux disease, mosapride, proton-pump inhibitor, treatment effectiveness

## Abstract

We aimed to evaluate whether adding a sustained-release (SR) formula of mosapride to proton-pump inhibitors (PPIs) would be more effective in controlling symptoms than PPI alone in patients with gastroesophageal reflux disease (GERD). Sixty patients with heartburn and/or regurgitation were randomly assigned to two groups: mosapride SR 15 mg combined with esomeprazole 20 mg once daily (ME group) and esomeprazole 20 mg once daily alone (E group). The primary endpoint was the complete-resolution rate of GERD symptoms after eight-week medication, and the secondary endpoints were the complete-resolution rate of GERD symptoms after four-week medication, symptom-improvement rates ≥ 50% after four- and eight-week medication, and change in reflux-disease-questionnaire (RDQ) and GERD-health-related quality-of-life (GERD-HRQL) scores from baseline at four- and eight-week medication. No significant differences in complete-symptom-resolution rates at eight weeks and four weeks or in the changes in RDQ and GERD-HRQL scores from baseline at four- and eight-week medication were observed between the ME and E groups. The symptom-improvement rate of ≥50% after four and eight weeks was comparable between both groups. Adding mosapride SR to esomeprazole in patients with GERD provides no additional benefits in controlling GERD symptoms.

## 1. Introduction

Gastroesophageal reflux disease (GERD) is defined as a digestive disorder that develops when the reflux of gastric contents causes troublesome symptoms, with or without visible damage to the esophageal mucosa [1]. Typical symptoms of GERD include heartburn, regurgitation, or both. The estimated worldwide prevalence of GERD is approximately 14.8% [2], and recent epidemiological studies have reported an increased incidence of GERD in Asia, particularly in Korea and Japan [3]. GERD can be divided into erosive reflux disease (ERD) and non-erosive reflux disease (NERD). Quality of life (QoL) is commonly impaired in patients with GERD, independent of the presence of esophageal lesions [4]. Proton-pump inhibitors (PPIs), which are antisecretory compounds, are typically used to treat GERD [5]. PPIs have been reported to provide symptomatic relief, heal erosion, and safely and effectively improve the QoL of patients with GERD [6,7]. Studies in patients with ERD treated with PPI once daily have reported 80–90% healing rates after eight weeks of therapy, with comparable symptom relief [8,9]. However, this effectiveness is approximately 20–30% lower in patients with NERD than in those with ERD because of a different pathophysiological pathway [10]. Moreover, approximately 30% of patients with GERD continue to manifest symptoms after treatment with standard doses of PPIs [10]. Consequently, other adjunctive treatments, including prokinetics, have been prescribed to improve the therapeutic effect of PPIs in clinical practice. Prokinetic agents have been theorized to be effective against GERD because they could enhance lower esophageal sphincter pressure and improve esophageal peristalsis, esophageal acid clearance, and gastric emptying [11].

Mosapride, a 5-hydroxytryptamine 4 (5-HT4) receptor agonist, is a prokinetic agent that can be safely used in patients with various gastrointestinal disorders [12]. It acts by increasing acetylcholine release from parasympathetic nerve endings and stimulating esophageal motility and gastric emptying [13,14]. Several previous studies have reported that combining mosapride with PPI was more effective than PPI alone in providing symptomatic relief to patients [15,16,17,18]. However, there is a limitation in prescribing mosapride to patients with GERD in real practice; although patients take PPI only once before breakfast, patients should take mosapride three times before meals, leading to problems of low compliance due to discomfort. Recently, a sustained-release (SR) formula for mosapride, which is released at a constant rate over the whole day, has been developed to increase patient compliance with medication. Mosapride SR has been reported to be as effective as conventional mosapride release three times daily in patients with functional dyspepsia [19]. However, the benefits of the mosapride-SR addition to PPIs in patients with GERD have not been evaluated. Therefore, this study aimed to investigate the additional effect of the new formula of mosapride over PPI alone in controlling GERD symptoms.

## 2. Materials and Methods

### 2.1. Study Population

This prospective, open-label, randomized controlled trial was conducted at Pusan National University Hospital (Busan, Korea) from July 2019 to October 2020. We enrolled patients aged ≥ 18 years that had typical GERD symptoms (heartburn and/or regurgitation) at least twice a week. Patients were excluded if they met the following criteria: (1) having symptoms indicative of severe or malignant diseases, including unintended weight loss, hematemesis, hematochezia, or jaundice; (2) diagnosed with pyloric stenosis, duodenal or gastric ulcer, Barrett’s esophagus (>3 cm in length), esophageal varix, gastrointestinal hemorrhage, or malignant diseases of the upper gastrointestinal tract based on esophagogastroduodenoscopy within last 3 months; (3) diagnosed with primary esophageal motility disorders, esophageal stricture, pancreatitis, disorders of absorption, inflammatory bowel diseases (such as Crohn’s disease or ulcerative colitis), or irritable bowel syndrome; (4) diagnosed with Zollinger–Ellison syndrome; (5) diagnosed with eosinophilic esophagitis; (6) diagnosed with severe pulmonary diseases within the last 3 months; (7) diagnosed with severe liver dysfunction or liver diseases; (8) diagnosed with severe renal diseases, including chronic renal disease or renal dysfunction; (9) diagnosed with uncontrolled diabetes or cerebrovascular diseases; (10) diagnosed within the last 3 months with a disease that requires surgery during the clinical-trial period; (11) diagnosed with malignant diseases within the last 5 years; (12) on drugs or alcohol abuse; (13) taking PPIs within the last 28 days, or histamine-2 receptor antagonists, sucralfate, prokinetics, or antacids daily up to the initial visit; and (14) pregnant or breastfeeding women, as well as female patients who were not willing to use contraception for the duration of the clinical-trial period.

The trial was conducted in accordance with the principles of good clinical practice and the Declaration of Helsinki. The study protocol was approved by the Korean Ministry of Food and Drug Safety and Institutional Review Board of Pusan National University Hospital (IRB number: 1809-008-085). Informed consent was obtained from all the participants at the time of enrollment. This trial was registered as a standard randomized clinical trial (cris.nih.go.kr: KCT0004062).

### 2.2. Randomization

During the screening visit, data on demographics, such as date of birth, sex, weight, height, and smoking and drinking habits were collected. Esophagogastroduodenoscopy was conducted in all patients within 3 months prior to enrollment. After obtaining informed consent, eligible patients were randomized into two treatment groups using computer-generated random numbers: (i) the ME group: mosapride citrate SR (Gasmotin SR^®^; Daewoong Pharmaceutical, Seoul, Korea) 15 mg and esomeprazole (Nexium^®^; Astrazeneca Korea, Seoul, Korea) 20 mg once daily before breakfast; and (ii) the E group: 20 mg esomeprazole alone once daily before breakfast. Each patient visited the hospital for evaluation of reflux symptoms 4 and 8 weeks after initiating the medication. Compliance was determined by the number of tablets remaining per drug type at the follow-up visit. If drug compliance was ≥80%, the patient data were included in the final outcome measurements.

### 2.3. Study Assessments

#### 2.3.1. GERD Symptom Assessment

To establish the severity of GERD symptoms, patients were asked to complete two validated GERD-specific instruments: (a) a reflux-disease questionnaire (RDQ) and (b) GERD health-related quality of life (GERD-HRQL). The RDQ is a 12-item questionnaire that was designed to assess the frequency and severity of heartburn (four items measuring the frequency and severity of pain and burning behind the breastbone), regurgitation (four items measuring the frequency and severity of acid taste in the mouth and movement of the material upward from the stomach), and dyspeptic complaints (four items measuring the frequency and severity of pain or burning in the upper stomach) [20]. Response options range from 0 (not present) to 5 (severe). The final score for each symptom was obtained by multiplying the scores for severity and frequency. The total score was obtained by adding the final scores for the individual symptoms. The GERD-HRQL was designed and validated to evaluate typical GERD symptoms by measuring ten items (six related to heartburn, two to dysphagia, one to bloating, and one to the impact of medication on daily life) on a visual analog scale ranging from 0 (no symptoms) to 5 (worst symptoms) [21]. The total score is calculated by summing the individual scores. After 4 and 8 weeks of treatment, all the participants completed the questionnaire to determine whether they achieved sufficient improvement in their GERD symptoms. Based on a questionnaire regarding GERD symptoms, the rates of improvement for the total scores and individual sub-scores were calculated.

#### 2.3.2. Endoscopic Assessment

The presence or absence of reflux esophagitis, hiatal hernia, and atrophic gastritis was determined during endoscopic examination. If esophagitis was present, it was graded according to the Los Angeles classification system [22]. A hiatal hernia was defined as a circular extension of the gastric mucosa above the diaphragmatic hiatus that was >2 cm in axial length. The atrophic gastritis grade was assessed endoscopically using the atrophic pattern system described by Kimura et al. [23]. This classification system divides the extent of atrophy into closed and open types. In the closed type, the atrophic border remains on the lesser curvature of the stomach, while in the open type, the atrophic border no longer exists on the lesser curvature but extends along the anterior and posterior walls of the stomach. Gastric antral and corpus biopsy samples were taken for the detection of *Helicobacter pylori* infection using a rapid urease test.

#### 2.3.3. Study Efficacy

The primary efficacy endpoint was the complete-resolution rate of GERD symptoms after 8-week medication. The secondary efficacy endpoints included the complete-resolution rate of GERD symptoms after 4-week medication, improvement rates of GERD symptoms at 4- and 8-week medication, and changes in RDQ and GERD-HRQL scores from baseline at 4- and 8-week medication. The complete-resolution rates of GERD symptoms were defined as when the RDQ total score reached zero after treatment at 4- and 8-week medication, and the improvement rate of GERD symptoms was defined as ≥50% reduction in the initial RDQ and GERD-HRQL scores at 4- and 8-week medication. The safety of medications was assessed by recording adverse events, including their severity and duration.

### 2.4. Sample Size and Statistical Analysis

The sample size was determined using data from a previous report that indicated complete relief from heartburn was achieved in 46.7% of patients treated with esomeprazole, and in 84.6% of those treated with tegaserod combined with esomeprazole [24]. The number of participants to detect a significant association with 80% power and a 5% two-sided type I error level (α = 0.05) was calculated to be 30 in each group, assuming a dropout rate of 20%.

The patient data were subjected to two types of analyses: intention-to-treat (ITT) and per-protocol (PP) sets. The ITT-set analysis included all patients who had data on the primary-endpoint-evaluation parameters after treatment with the clinical-trial drugs. The PP-set analysis focused on patients from the ITT analysis with data indicating that these patients had completed the clinical trial according to the protocol. Safety analysis included all data from randomly assigned patients who took the study drugs. Efficacy parameters are presented as frequency and proportion (95% confidence interval [CI]) in each group. Statistical analyses were performed using the t-test for parametric data, Mann–Whitney *U*-test for non-parametric data, and χ^2^ test or Fisher’s exact test for categorical data. All statistical analyses were performed using SAS, version 9.4 (SAS Institute, Cary, NC, USA). A two-sided *p* value of ≤0.05 was considered significant.

## 3. Results

### 3.1. Allocation of Patients and Baseline Clinicodemographic Characteristics

Altogether, 60 patients were included. Among these patients, 30 were randomized to the E group and 30 to the ME group. One patient in the E group and four in the ME group were excluded from the ITT analysis because of consent withdrawal owing to insufficient satisfaction (Figure 1). Consequently, the data of 55 patients (E group, n = 29; ME group, n = 26) were used in the PP analysis.

Table 1 lists the baseline clinicodemographic characteristics of all the patients. No differences in age, sex, body-mass index, smoking status, alcohol consumption, or GERD-symptom scores were observed between the two groups (Table 1). The baseline endoscopic findings (reflux esophagitis, hiatal hernia, and atrophic gastritis) were also comparable between the two groups. Drug compliance rates ≥80% throughout the treatment period were reported by all participants.

### 3.2. Efficacy Assessment

#### 3.2.1. Complete-Resolution Rates of GERD Symptoms at Four- and Eight-Week Medication

Based on the ITT analysis, the complete-resolution rates of GERD symptoms eight weeks after treatment initiation were 43.3% (13/30) and 23.3% (7/30) in the E and ME groups, respectively, and no significant difference in the complete-resolution rates of GERD symptoms was observed between both groups (*p* = 0.171) (Table 2). In the PP analysis, the complete-resolution rates of GERD symptoms eight weeks after treatment initiation were 44.8% (13/29) and 26.9% (7/26) in the E and ME groups, respectively (*p* = 0.272).

After four weeks from treatment initiation, GERD symptoms completely resolved in only three patients in the E group. Based on the ITT analysis, the complete-resolution rates of GERD symptoms were 10.0% (3/30) and 0% (0/30) in the E and ME groups, respectively; no significant difference was identified between both groups (*p* = 0.236). In the PP analysis, the complete-resolution rates of GERD symptoms were 10.3% (3/29) and 0% (0/26) in the E and ME groups, respectively (*p* = 0.275).

#### 3.2.2. Improvement Rates of GERD Symptoms at Four- and Eight-Week Medication

In the ITT analysis, at eight weeks of medication, 80% of patients (24/30) in the E group and 76.7% of patients (23/30) in the ME group demonstrated a ≥50% reduction in the RDQ total score; no significant difference in the improvement rates of RDQ total score was observed between the two groups (*p* = 1.000) (Table 3). Additionally, no significant differences in the subscales of heartburn, regurgitation, and dyspeptic symptoms were observed between the E and ME groups (70.0% vs. 73.3%, *p* = 1.000; 73.3% vs. 60.0%, *p* = 0.637; and 73.3% vs. 60.0%, *p* = 0.092). At four weeks of treatment, no differences in the RDQ total symptom score and each symptom score were identified between the two groups. These results were similar to those of the PP analysis.

#### 3.2.3. Change of RDQ and GERD-HRQL Scores from Baseline at Four- and Eight-Week Medication

(1)Change of RDQ scores after four- and eight-week medication
Both treatment groups achieved significant decreases in RDQ total scores relative to the baseline after four weeks (frequency: E group −2.37 ± 3.52, *p* = 0.001 and ME group −1.77 ± 5.35, *p* = 0.081; severity: E group −3.83 ± 4.93, *p* < 0.001 and ME group −2.50 ± 4.78, *p* < 0.001) and after eight weeks of therapy (frequency: E group −7.07 ± 5.43, *p* < 0.001; ME group −6.17 ± 5.61, *p* < 0.001; severity: E group −8.77 ± 6.48, *p* < 0.001 and ME group −6.70 ± 3.75, *p* < 0.001). Between the two groups, no significant difference was observed in changes in the frequency (four weeks, *p* = 0.610 and eight weeks, *p* = 0.531) and severity (four weeks, *p* = 0.292 and eight weeks, *p* = 0.137) of GERD symptoms (Table 4).

Heartburn scores significantly decreased after eight weeks of treatment in both groups. However, at four weeks after medication, the E group demonstrated a significant decrease in heartburn scores (frequency, *p* = 0.032; severity, *p* = 0.015), while the ME group did not demonstrate a significant decrease in heartburn scores (frequency, *p* = 0.092; severity, *p* = 0.291). Between the two groups, no significant difference was identified in the changes in the frequency (four weeks, *p* = 0.518 and eight weeks, *p* = 0.309) and severity (four weeks, *p* = 0.749 and eight weeks, *p* = 0.423) of heartburn.

Regurgitation scores significantly decreased at four and eight weeks after treatment in both the treatment groups. Between the two groups, no significant difference was observed in the changes in the frequency (four weeks, *p* = 0.554 and eight weeks, *p* = 0.579) and severity (four weeks, *p* = 1.000 and eight weeks, *p* = 0.844) of regurgitation.

Dyspeptic-symptom scores significantly decreased after eight weeks of treatment in both groups. However, at four weeks after medication, only the severity of dyspepsia in the E group was significantly decreased, and other components did not change significantly (frequency: E group, *p* = 0.152 and ME group, *p* = 0.629; severity: E group, *p* = 0.003 and ME group, *p* = 0.074). Between the two groups, no significant difference was observed in the changes in the frequency (four weeks, *p* = 0.475 and eight weeks, *p* = 0.949) and severity (four weeks, *p* = 0.489 and eight weeks, *p* = 0.188) of dyspeptic symptoms.

(2)Change of GERD-HRQL Scores After Four- and Eight-Week Medication
After four weeks of medication, both treatment groups demonstrated decreased GERD-HRQL scores compared to those at baseline, although this difference was not significant (E group −1.80 ± 6.19, *p* = 0.122 and ME group −2.43 ± 7.12, *p* = 0.071). After eight weeks of medication, both groups achieved a significant decrease in GERD-HRQL scores (E group −7.10 ± 6.50, *p* < 0.001; ME group −5.57 ± 4.55, *p* < 0.001). Between the two groups, no significant difference was observed in the changes in the GERD-HRQL scores from baseline to four- and eight-weeks (four weeks, *p* = 0.715 and eight weeks, *p* = 0.295).

### 3.3. Adverse Events

During the study period, four patients in the E group (13.3%, seven cases) and six patients in the ME group (20%, ten cases) reported adverse events (Table 5). Gastrointestinal symptoms were the most common events, and the occurrence of adverse events did not vary significantly between the two groups (*p* = 0.731). No serious adverse events were recorded.

## 4. Discussion

In this randomized controlled trial evaluating the efficacy of combining mosapride SR with esomeprazole in patients with GERD compared with esomeprazole monotherapy, GERD symptoms were significantly improved compared to the baseline in both the ME and E groups, as demonstrated by the reduction in the RDQ and GERD-HRQL scores after the four- and eight-week medication periods. However, mosapride SR over esomeprazole alone did not provide additional benefits in controlling the GERD symptoms.

In a previous study using esomeprazole in patients with GERD, the complete-response rate was 60.0% for heartburn and 61.8% for regurgitation after an eight-week treatment [25], whereas in this study, it was only 33.3% for total GERD symptoms based on the RDQ scores. The relatively lower response rate might be explained by the difference in the heterogeneity of the baseline clinicodemographic background, such as a higher proportion of patients with mild GERD symptoms and those with NERD, the difference in the questionnaire used for GERD symptoms, and the difference in dosage used for esomeprazole (40 mg vs. 20 mg). Here, we selected half-dose (20 mg) esomeprazole rather than full-dose (40 mg) esomeprazole in order to better confirm the additional effect of mosapride SR on PPI, as well as real-world insurance regulations. In Korea, half-dose PPI is covered by government insurance.

In this study, mosapride SR did not demonstrate additional benefits in terms of both the complete-resolution and improvement rates of GERD symptoms in the patients. These results are consistent with those of other randomized controlled trials [26,27,28,29] and a systemic review [30]. Yamaji et al. conducted a randomized controlled trial comparing the efficacy of mosapride-plus-omeprazole combination therapy to omeprazole monotherapy in patients with GERD, and concluded no additional benefit of combining mosapride with PPI for treating reflux symptoms compared to PPI alone [28]. In a systematic review regarding the potential benefits of mosapride plus PPI for treating GERD, the mosapride combined therapy was not more effective than PPI alone as a first-line therapy [30]. By contrast, a randomized controlled trial including only patients with ERD has revealed that combination therapy with esomeprazole and mosapride would be useful for rapid symptom relief [16]. Madan et al. have reported that the combination of mosapride and pantoprazole significantly improved symptom control in patients with ERD, but not in those with NERD [17]. In a recent meta-analysis, mosapride combined with PPI significantly improved the reflux-symptom score compared to PPI alone, specifically in patients with ERD [18]. The difference in the results might be attributed to the different proportions of patients with NERD in these studies. PPI is widely known to be less effective for treating NERD [10,31]. A systematic review of the literature has revealed that the PPI symptomatic-response pooled rate was only 36.7% in patients with NERD [10]. Based on these findings, symptom improvement with the addition of mosapride would be particularly expected in patients with NERD [32]. However, several studies have reported that the benefit of mosapride addition is more dominant in patients with ERD than in those with NERD. In the present study, there was no difference in GERD-symptom improvement between the combination and PPI monotherapy, even in patients with ERD, although the number of such patients was small.

Several studies have also suggested that patients with certain specific symptoms might be responsive to the addition of mosapride to PPI; specifically, PPI-plus-mosapride combination therapy might be effective in selected patients, but not all patients with GERD. The addition of mosapride was more beneficial in patients with burping [28], severe GERD symptoms before treatment [33], and PPI-resistant NERD with delayed gastric emptying [14].

Mosapride, a 5-HT4 receptor agonist, is a prokinetic drug used in functional gastrointestinal disorders, including GERD and functional dyspepsia, and is safe because it does not provoke QT prolongation [34]. The mechanism of action of mosapride includes the enhancement of esophageal motor function, acceleration of gastric emptying, and enhanced acid-inhibitory effect of PPIs in humans [13]. In clinical practice, physicians frequently prescribe adjunctive therapy to PPI in patients with GERD, especially prokinetics such as mosapride. However, in contrast to PPI, mosapride is taken three times a day, which leads to treatment non-adherence. Mosapride SR contains two components: a rapid-release component and a slow-release component. Mosapride SR once a day has pharmacokinetic characteristics similar to those of conventional mosapride administered three times a day. In a previous study comparing the efficacy of mosapride SR and conventional mosapride in patients with functional dyspepsia, mosapride SR once daily was as effective as conventional mosapride three times daily, with a similar safety profile based on patient-reported symptom improvement and dyspepsia-specific QoL [19]. However, the effect of mosapride-SR addition to PPIs in patients with GERD has not been previously reported. To the best of our knowledge, this is the first study to evaluate the additional effect of mosapride SR on PPI use in patients with GERD. Although our study could not prove the significant benefits of mosapride-SR addition to PPI compared with PPI monotherapy in patients with GERD, mosapride SR can be added to the treatment of selected patients suggested in the aforementioned studies. Further, prospective, large-scale, multi-center studies are warranted to investigate subpopulations of patients with GERD in whom additional effects of mosapride SR may be helpful for symptom control.

Our study had several limitations. First, this was an open-label, randomized controlled trial, and not a blinded, placebo-controlled study. Second, the study population was heterogeneous in this study, the diagnosis of GERD was based on the patients’ symptoms, and the 24 h ambulatory pH/impedance-monitoring test was not performed; therefore, some patients with NERD might not have had real reflux diseases, such as reflux hypersensitivity or functional heartburn. For a better study population, accurately diagnosing NERD is crucial; however, in clinical practice, NERD is generally diagnosed based on reflux symptoms without abnormal endoscopic findings in the esophagus [35]. Accordingly, the relatively lower complete-resolution rate of GERD symptoms in the ME group could be explained to some degree by the possibility that more patients with reflux hypersensitivity or functional heartburn were included in the ME group. Third, this study had a relatively small number of patients available for sub-analysis to identify specific situations of patients who benefit from combination therapy over PPI monotherapy.

In conclusion, the addition of mosapride SR to esomeprazole in patients with GERD did not provide additional benefits in controlling GERD symptoms. However, considering the usefulness of conventional mosapride in patients with GERD, prospective, large-scale, multi-center studies are needed to elucidate the subpopulation of patients with GERD in whom additional effects of mosapride SR are helpful for symptom control.

## Figures and Tables

**Figure 1 jcm-11-01965-f001:**
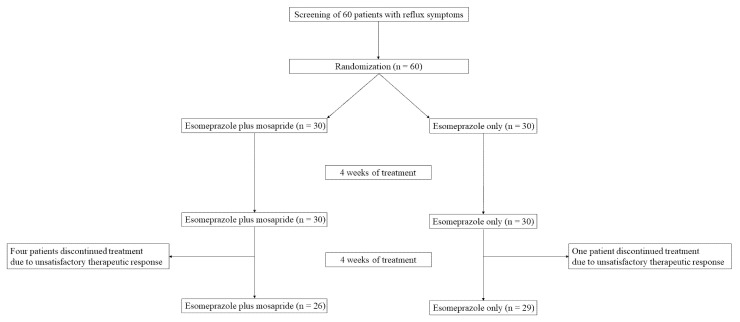
Flow chart of the patients included in the study.

**Table 1 jcm-11-01965-t001:** Baseline clinicodemographic characteristics of the study patients.

	Esomeprazole	Esomeprazole + Mosapride	*p* Value
	(n = 30)	(n = 30)	
Age, years	34.7 ± 13.2	38.2 ± 14.0	0.323
Sex			0.381
Man	6 (20.0)	10 (33.3)	
Woman	24 (80.0)	20 (66.7)	
Height, cm	163.2 ± 7.3	163.2 ± 10.3	1.000
Body weight, kg	59.5 ± 10.3	60.6 ± 10.6	0.685
Body-mass index, kg/m^2^	22.2 ± 7.0	22.9 ± 7.3	0.685
Waist, cm	73.1 ± 7.9	72.9 ± 10.6	0.916
Alcohol drinking	26 (86.7)	26 (86.7)	1.000
Smoking	2 (6.7)	7 (23.3)	0.148
GERD-symptom scores			
RDQ score	36.7 ± 21.1	31.3 ± 17.0	0.285
GERD-HRQL score	15.8 ± 7.6	14.7 ± 6.3	0.543
Reflux esophagitis			1.000
Absent	23 (76.6)	22 (73.3)	
Present	7 (23.3)	8 (26.7)	
Hiatal hernia			1.000
Absent	28 (93.3)	29 (96.7)	
Present	2 (6.7)	1 (3.3)	
Atrophic gastritis			0.731
Absent	6 (20.0)	4 (13.3)	
Closed type	24 (80.0)	25 (83.3)	
Open type	0	1 (3.3)	
*Helicobacter pylori* infection			0.424
Absent	28 (93.3)	25 (83.3)	
Present	2 (6.7)	5 (16.7)	

Data are presented as mean ± standard deviation or number (%). GERD, gastroesophageal reflux disease; HRQL, health-related quality of life; RDQ, reflux-disease questionnaire.

**Table 2 jcm-11-01965-t002:** Complete-resolution rates of gastroesophageal reflux symptoms at four and eight weeks.

	Week 4			Week 8		
	Esomeprazole	Esomeprazole + Mosapride	*p* Value	Esomeprazole	Esomeprazole + Mosapride	*p* Value
Intention-to-treat set						
Number of patients	30	30		30	30	
Complete-resolution rate	3 (10.0)	0 (0.0)	0.236	13 (43.3)	7 (23.3)	0.171
Per-protocol set						
Number of patients	29	26		29	26	
Complete-resolution rate	3 (10.3)	0 (0.0)	0.275	13 (44.8)	7 (26.9)	0.272

Data are presented as number (%).

**Table 3 jcm-11-01965-t003:** Improvement rates of gastroesophageal reflux symptoms at four and eight weeks.

Symptom Improvement	Week 4			Week 8		
	Esomeprazole	Esomeprazole + Mosapride	*p* Value	Esomeprazole	Esomeprazole + Mosapride	*p* Value
Intention-to-treat set						
Number of patients	30	30		30	30	
Total symptom	11 (36.7)	14 (46.7)	0.600	24 (80.0)	23 (76.7)	1.000
Heartburn	2 (6.7)	3 (10.0)	1.000	21 (70.0)	22 (73.3)	1.000
Regurgitation	4 (13.3)	3 (10.0)	1.000	22 (73.3)	18 (60.0)	0.637
Dyspepsia	2 (6.7)	2 (6.9)	1.000	22 (73.3)	18 (60.0)	0.092
Per-protocol set						
Number of patients	29	26		29	26	
Total symptom	10 (34.5)	10 (38.5)	0.980	23 (79.3)	19 (73.1)	0.822
Heartburn	2 (6.9)	1 (3.8)	1.000	21 (72.4)	18 (62.1)	1.000
Regurgitation	4 (13.8)	1 (3.8)	0.506	22 (75.9)	14 (53.8)	0.279
Dyspepsia	2 (6.9)	1 (3.8)	1.000	21 (72.4)	15 (57.7)	0.096

Data are presented as number (%).

**Table 4 jcm-11-01965-t004:** Changes of reflux-disease questionnaire and gastroesophageal health-related quality of life scores from baseline at four- and eight-week medication.

	Baseline		Change from Baseline at Week 4			Change from Baseline at Week 8		
	Esomeprazole	Esomeprazole + Mosapride	Esomeprazole	Esomeprazole + Mosapride	*p* Value	Esomeprazole	Esomeprazole + Mosapride	*p* Value
	(n = 30)	(n = 30)	(n = 30)	(n = 30)		(n = 30)	(n = 30)	
RDQ score								
Total								
Frequency	12.50 ± 4.85	11.40 ± 5.36	−2.37 ± 3.52	−1.77 ± 5.35	0.610	−7.07 ± 5.43	−6.17 ± 5.61	0.531
Severity	13.53 ± 6.18	11.63 ± 4.25	−3.83 ± 4.93	−2.50 ± 4.78	0.292	−8.77 ± 6.48	−6.70 ± 3.75	0.137
Heartburn								
Frequency	3.60 ± 2.01	3.47 ± 2.33	−0.47 ± 1.31	−0.70 ± 1.47	0.518	−1.10 ± 1.37	−1.50 ± 1.59	0.309
Severity	3.90 ± 2.59	3.20 ± 1.77	−0.13 ± 1.55	0.00 ± 1.66	0.749	−0.90 ± 1.59	−0.57 ± 1.55	0.423
Regurgitation								
Frequency	4.70 ± 2.38	4.27 ± 2.30	−0.50 ± 1.33	−0.27 ± 1.68	0.554	−1.28 ± 1.60	−1.03 ± 1.73	0.579
Severity	4.87 ± 2.62	4.27 ± 2.55	−0.50 ± 1.31	−0.50 ± 1.41	1.000	−1.41 ± 1.55	−1.33 ± 1.58	0.844
Dyspepsia								
Frequency	4.20 ± 2.22	3.67 ± 2.31	−0.23 ± 0.97	0.00 ± 1.49	0.475	−0.76 ± 1.33	−0.73 ± 1.68	0.949
Severity	4.77 ± 2.40	4.17 ± 2.36	−0.53 ± 1.20	−0.30 ± 1.39	0.489	−1.48 ± 1.09	−1.00 ± 1.64	0.188
GERD-HRQL score	14.87 ± 7.43	13.73 ± 6.27	−1.80 ± 6.19	−2.43 ± 7.12	0.715	−7.10 ± 6.50	−5.57 ± 4.55	0.295

Data are presented as mean ± standard deviation.

**Table 5 jcm-11-01965-t005:** Incidence of adverse drug reactions of the two medications.

	Overall	Esomeprazole	Esomeprazole + Mosapride
Epigastric soreness	4	2	2
Indigestion	3	0	3
Constipation	3	2	1
Abdominal pain	1	0	1
Diarrhea	1	0	1
Abdominal bloating	1	0	1
Palpitation	1	0	1
Chest discomfort	1	1	0
Sore taste	1	1	0
Regurgitation	1	1	0
Total	17	7	10

## Data Availability

The data presented in this study are available on request from the corresponding author.

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
