# Peer review of "Randomized Controlled Trial Comparing the Efficacy of Sustained-Release Formula of Mosapride-Plus-Esomeprazole Combination Therapy to Esomeprazole Monotherapy in Patients with Gastroesophageal Reflux Disease"

_jcm, 2022, doi:10.3390/jcm11071965_

Round 1
Reviewer 1 Report
This study aimed to compare the additional effect of mosapride SR over PPI alone in controlling GERD symptoms. As the authors mentioned, mosapride SR did not demonstrate additional benefits in terms of both the complete resolution and improvement rates of GERD symptoms in the included patients. Their results are consistent with those of other randomized controlled trials. Congratulations to your data. However, I still have a few questions as the following.
First, regarding your strict exclusion/incision criteria and the short time frame of study period (July 2019 to October 2020), please add the different kind of dropout in the figure 1 to better demonstrate the process of patient recruitment.
Second, how did you deliver the medication to the patients in separated arms, one with two drugs and the another with single drug? As you claimed that this is a randomized trial but without blindness, can you explain more specifically at this point?
Third, I am wondering if any conflict of interest in your study? As you mentioned this study was supported by Daewoong Pharmaceutical Company, Ltd. (Seoul, Korea), which also produced the drug (mosapride citrate SR). Please clarify this issue.
Author Response
# Reply to Reviewer 1’s comments
We would like to thank the reviewer for the constructive critique to improve the manuscript. We have made every effort to address the issues raised and to respond to all comments. The revisions are indicated in red font in the revised manuscript. Please find below a detailed, point-by-point response to the reviewer's comments. We hope that our revisions meet the reviewer’s expectations.
- First, regarding your strict exclusion/incision criteria and the short time frame of study period (July 2019 to October 2020), please add the different kind of dropout in the figure 1 to better demonstrate the process of patient recruitment.
- We appreciate your valuable comment. All the 60 patients subjected to screening were randomly allocated into groups, with 30 in the ME group and 30 in the E only group. According to the Reviewer’s suggestion, we revised Fig. 1.
Figure 1. Flow chart of the patients included in the study.
- Second, how did you deliver the medication to the patients in separated arms, one with two drugs and the another with single drug? As you claimed that this is a randomized trial but without blindness, can you explain more specifically at this point?
- As mentioned, this was an open-label, randomized controlled trial, but not a placebo-controlled study. Therefore, after randomization, participants could be aware of their treatment arm. They received each medicine through clinical pharmacies of the Pusan National University Hospital. We have already described this issue in the Discussion section.
Our study had several limitations. First, this was an open-label, randomized controlled trial, and not a blinded, placebo-controlled study.
- Third, I am wondering if any conflict of interest in your study? As you mentioned this study was supported by Daewoong Pharmaceutical Company, Ltd. (Seoul, Korea), which also produced the drug (mosapride citrate SR). Please clarify this issue.
- As stated in the Acknowledgments section, this study was supported by Daewoong Pharmaceutical Company, Ltd. (Seoul, Korea). As the Reviewer pointed out, mosapride citrate SR is produced by this company. Therefore, we have revised “Conflicts of Interest” statement.
Conflicts of Interest: The corresponding author, Kim GH, received funds from Daewoong Pharmaceutical Company, Ltd. All other authors declare no conflict of interest regarding this study.
Reviewer 2 Report
I would like to congratulate the authors on their research and the well written manuscript.
A few comments:
- In the discussion, I would like the authors to discuss is why their results show a signal that the ME group is much less effective than the E group in the complete resolution of symptoms ( about a 20% difference)
- Since over 70% of patients had erosive esophagitis ( reflux esophagitis)- could they do a sub-group analysis for the outcomes just for that sub-group.
Author Response
# Reply to Reviewer 2’s comments
We would like to thank the reviewer for the constructive critique to improve the manuscript. We have made every effort to address the issues raised and to respond to all comments. The revisions are indicated in red font in the revised manuscript. Please find below a detailed, point-by-point response to the reviewer's comments. We hope that our revisions meet the reviewer’s expectations.
- In the discussion, I would like the authors to discuss is why their results show a signal that the ME group is much less effective than the E group in the complete resolution of symptoms (about a 20% difference)
- As the Reviewer pointed out, the complete resolution rates of GERD symptoms in the E and ME groups were 43.3% (13/30) and 23.3%, respectively. However, the difference between both groups was not statistically significant (p = 0.171). This could be explained by a relatively small number of included patients and the heterogeneity of the study population. Especially, the proportion of NERD patients was approximately 75%; considering that NERD consists of three categories (true NERD, reflux hypersensitivity, and functional heartburn), the relatively lower complete resolution rates of GERD symptoms could be explained to some degree by a possibility that the proportion of patients with reflux hypersensitivity and functional heartburn was higher in the ME group than that in the E group. Accordingly, we have added the following text in the Discussion section.
Second, the study population was heterogeneous in this study, the diagnosis of GERD was based on the patients’ symptoms, and the 24-hour ambulatory pH/impedance monitoring test was not performed; therefore, some patients with NERD might not have had real reflux diseases, such as reflux hypersensitivity or functional heartburn. For a better study population, accurately diagnosing NERD is crucial; however, in clinical practice, NERD is generally diagnosed based on reflux symptoms without abnormal endoscopic findings in the esophagus [35]. Accordingly, the relatively lower complete resolution rate of GERD symptoms in the ME group could be explained to some degree by a possibility that more patients with reflux hypersensitivity or functional heartburn were included in the ME group.
- Since over 70% of patients had erosive esophagitis (reflux esophagitis)- could they do a sub-group analysis for the outcomes just for that sub-group.
- The proportion of erosive esophagitis were 23.3% (7/30) and 26.7% (8/30) in the E and ME groups, respectively. We were not able to perform a sub-group analysis due to the small number of patients. We sincerely request the understanding of the Reviewer.
Reviewer 3 Report
Good designed prospective study , however the number of patients inckuded in each group is low. This is a limitation. Could you increase the number of paatients in each group? you can wait for more time and so obtein more sustancial results.
Have you endoscopic, manometric and 24hr ph monitoring evaluation before and after treatment. These kind of study also can improve your study
Author Response
# Reply to Reviewer 3’s comments
We would like to thank the reviewer for the constructive critique to improve the manuscript. We have made every effort to address the issues raised and to respond to all comments. The revisions are indicated in red font in the revised manuscript. Please find below a detailed, point-by-point response to the reviewer's comments. We hope that our revisions meet the reviewer’s expectations.
- Good designed prospective study, however, the number of patients included in each group is low. This is a limitation. Could you increase the number of patients in each group? you can wait for more time and so obtain more substantial results.
- As stated in the Sample size and Statistical Analysis sections, we calculated the sample size based on a previous report that complete relief from heartburn was achieved in 46.7% of patients treated with esomeprazole and in 84.6% of those treated with tegaserod combined with esomeprazole. Accordingly, the number of participants to detect a significant association with 80% power and a 5% two-sided type I error level (α = 05) was calculated to be 30 in each group, assuming a dropout rate of 20 %.
- We do agree with the Reviewer thought. If we get more funds in the future, we will include more patients. However, at the present time, we do not have enough budget. We sincerely request the understanding of the Reviewer.
- Have you endoscopic, manometric and 24hr ph monitoring evaluation before and after treatment? This kind of study also can improve your study.
- We completely agree with the Reviewer’s comments. We have also described this limitation in the Discussion section. As aforementioned, if we get enough budget, we will perform another prospective study that will include endoscopy, high-resolution manometry, and 24-hour impedance-pH monitoring before/after the medication to evaluate the additive effect of mosapride in patients with true GERD.
Reviewer 4 Report
Abstract:
The abstract is unstructured. It is missing to separate in: 1) Background, 2) Methods, 3) Results, and 4) Conclusion.
Introduction:
It is well structured and very understandable. Justifies the hypothesis of the study. In addition, the bibliography presented is relatively recent.
Material and methods:
A more precise indication of the structure / composition of sustained release mosapride would be nice. a part of this is explained in the discussion lines 311 to 317.
The primary and secondary objectives are described in the abstract but not in the material and methods section. They should be indicated in this section.
Study population: Definition of typical GERD. Indicate the bibliographic citation of this description.
Endoscopic assessment: Indicate the number of samples obtained in the endoscopic examination (mean and SD)
Results :
Table 1 : It is assumed that this is a randomized study. Therefore, if the randomization is correct, there should be no statistical differences. It is not necessary to indicate the p values in the table.
Table 5: It takes up too much space in the article and can be summarized with a written description
Discussion:
- The opening sentences of the discussion should explain the result of the primary objective, which in this case is not significant. In the articles, a secondary objective is first exposed, which is significant, but in any case, it should be highlighted later.
Author Response
# Reply to Reviewer 4’s comments
We would like to thank the reviewer for the constructive critique to improve the manuscript. We have made every effort to address the issues raised and to respond to all comments. The revisions are indicated in red font in the revised manuscript. Please find below a detailed, point-by-point response to the reviewer's comments. We hope that our revisions meet the reviewer’s expectations.
- Abstract: The abstract is unstructured. It is missing to separate in: 1) Background, 2) Methods, 3) Results, and 4) Conclusion.
- We removed the subheadings in the Abstract section to ensure conformance to the guidelines of the target journal, Journal of Clinical Medicine.
- Introduction: It is well structured and very understandable. Justifies the hypothesis of the study. In addition, the bibliography presented is relatively recent.
- We appreciate the valuable and encouraging comments of the Reviewer.
- Material and methods: A more precise indication of the structure/composition of sustained release mosapride would be nice. a part of this is explained in the discussion lines 311 to 317.
- We have described the structure and indications of mosapride and mosapride SR in the Introduction. Therefore, we would like to maintain the current form.
Mosapride, a 5-hydroxytryptamine 4 (5-HT4) receptor agonist, is a prokinetic agent that can be safely used in patients with various gastrointestinal disorders 1. It acts by increasing acetylcholine release from parasympathetic nerve endings and stimulating esophageal motility and gastric emptying 2,3. Several previous studies have reported that combining mosapride with PPI was more effective than PPI alone in providing symptomatic relief to patients 4-7. However, there is a limitation in prescribing mosapride to patients with GERD in real practice; although patients take PPI only once before breakfast, patients should take mosapride three times before meals, leading to problems of low compliance due to discomfort. Recently, a sustained release (SR) formula for mosapride, which is released at a constant rate over the whole day, has been developed to increase patient compliance with medication. Mosapride SR has been reported to be as effective as conventional mosapride release three times daily in patients with functional dyspepsia.
- Material and methods: The primary and secondary objectives are described in the abstract but not in the material and methods section. They should be indicated in this section.
- We have already described the primary and secondary efficacy endpoints in section 2.3.3., the Study Efficacy section.
2.3.3. Study Efficacy
The primary efficacy endpoint was the complete resolution rate of GERD symptoms after 8-week medication. The secondary efficacy endpoints included the complete resolution rate of GERD symptoms after 4-week medication, improvement rates of GERD symptoms at 4- and 8-week medication, and changes in RDQ and GERD-HRQL scores from baseline at 4- and 8-week medication. The complete resolution rates of GERD symptoms were defined as when the RDQ total score reached zero after treatment at 4- and 8-week medication, and the improvement rate of GERD symptoms was defined as ≥50% reduction in the initial RDQ and GERD-HRQL scores at 4- and 8-week medication. The safety of medications was assessed by recording adverse events, including their severity and duration.
- Study population: Definition of typical GERD. Indicate the bibliographic citation of this description.
- It is well known that heartburn and/or regurgitation are typical GERD symptoms. In fact, most articles also use this definition without citation. Therefore, we do not need to insert the bibliographic citation. We sincerely request the understanding of the Reviewer.
- Endoscopic assessment: Indicate the number of samples obtained in the endoscopic examination (mean and SD).
- We have already described the number of endoscopies in section 2.2., the Randomization section. In addition, the endoscopic findings are non-continuous variables, therefore, they are expressed as number (%) and not as mean and SD.
2.2. Randomization
During the screening visit, data on demographics, such as date of birth, sex, weight, height, and smoking and drinking habits were collected. Esophagogastroduodenoscopy was conducted in all patients within 3 months prior to enrollment.
- Table 1: It is assumed that this is a randomized study. Therefore, if the randomization is correct, there should be no statistical differences. It is not necessary to indicate the p values in the table.
- Although this is a randomized study, it is impossible to perform randomization based on all parameters. Therefore, there is a possibility of bias in the paraments influencing the therapeutic effects such as the proportion of smoking, alcohol drinking, and pylori infection. To show that there were no differences in these parameters between the two groups, we inserted the p values in the Table.
- Table 5: It takes up too much space in the article and can be summarized with a written description.
- We would like to retain the Table 5 to apprise the readers regarding the adverse drug reactions in the two groups. We sincerely request the understanding of the Reviewer.
- Discussion: The opening sentences of the discussion should explain the result of the primary objective, which in this case is not significant. In the articles, a secondary objective is first exposed, which is significant, but in any case, it should be highlighted later.
- We appreciate the valuable comments of the Reviewer. However, we think that the current form of informing the primary objectives is more natural and effective. We sincerely request the understanding of the Reviewer.
Round 2
Reviewer 3 Report
accepted